# Phase Transitions and Structural Changes in DPPC Liposomes Induced by a 1-Carba-Alpha-Tocopherol Analogue

**DOI:** 10.3390/molecules26102851

**Published:** 2021-05-11

**Authors:** Grażyna Neunert, Jolanta Tomaszewska-Gras, Aneta Baj, Marlena Gauza-Włodarczyk, Stanislaw Witkowski, Krzysztof Polewski

**Affiliations:** 1Department of Physics and Biophysics, Faculty of Food Science and Nutrition, Poznan University of Life Sciences, Wojska Polskiego 38/42, 60-637 Poznan, Poland; grazyna.neunert@up.poznan.pl; 2Department of Food Quality and Safety Management, Faculty of Food Science and Nutrition, Poznan University of Life Sciences, Wojska Polskiego 31/33, 60-637 Poznan, Poland; jolanta.tomaszewska-gras@up.poznan.pl; 3Department of Organic Chemistry, Faculty of Chemistry, University of Bialystok, Ciolkowskiego 1K, 15-245 Bialystok, Poland; aneta.baj@uwb.edu.pl (A.B.); wit@uwb.edu.pl (S.W.); 4Department of Biophysics, Faculty of Medical Sciences, Poznan University of Medical Sciences, Grunwaldzka 6, 60-780 Poznan, Poland; mgauza@ump.edu.pl

**Keywords:** ANS fluorescence, DPH emission polarization, alpha-tocopherol carba analogue, phase transition, liposome melting

## Abstract

Steady-state emission spectroscopy of 1-anilino-8- naphthalene sulfonate (ANS) and 1,6-diphenyl-1,3,5-hexatriene (DPH), fluorescence anisotropy, and DSC methods were used to characterize the interactions of the newly synthesized 1-carba-alpha-tocopherol (CT) with a 1,2-dipalmitoyl-*sn*-glycero-3-phosphocholine (DPPC) membrane. The DSC results showed significant perturbations in the DPPC structure for CT concentrations as low as 2 mol%. The main phase transition peak was broadened and shifted to lower temperatures in a concentration-dependent manner, and pretransition was abolished. Increasing CT concentrations induced the formation of new phases in the DPPC structure, leading to melting at lower temperatures and, finally, disruption of the ordered DPPC structure. Hydration and structural changes of the DPPC liposomes using ANS and DPH fluorescent probes, which are selectively located at different places in the bilayer, were studied. With the increased concentration of CT molecules in the DPPC liposomes, structural changes with the simultaneous formation of different phases of such mixture were observed. Temperature studies of such mixtures revealed a decrease in the temperature of the main phase transition and fluidization at decreasing temperatures related to increasing hydration in the bilayer. Contour plots obtained from concentration–temperature data with fluorescent probes allowed for identification of different phases, such as gel, ordered liquid, disordered liquid, and liquid crystalline phases. The CT molecule with a modified chromanol ring embedded in the bilayer led to H-bonding interactions, expelling water molecules from the interphase, thus introducing disorder and structural changes to the highly ordered gel phase.

## 1. Introduction

Vitamin E (α-tocopherol) is regarded as one of the most effective natural lipophilic chain-breaking antioxidants, protecting the integrity of lipids and phospholipid bilayers of biomembranes [1]. It also exhibits a broad spectrum of nonantioxidant biological functions [2,3]. The most widely accepted biological function of vitamin E is its antioxidant activity [4]. This functionality is fulfilled by stereoelectronic features of the chroman-6-ol system which contributes to the stabilization of the tocopheroxyl radicals that are formed. Their stability arises from the overlapping of the *p*-type lone pairs of the oxygen atom O1 and the π-electrons of the fully substituted chroman-6-ol ring, rendering the formed phenoxy radical relatively unreactive [5,6,7]. In search of a new synthetic, more efficient tocopherol analogues and for better evaluation of the tocopheroxyl radical (TocO·) stabilization mechanism, a few tocopherol analogues have been synthesized. However, among them, the most interesting seems to be a new molecule of 1-carba-α-tocopherol (CT) since, in this molecule, the heterocyclic oxygen O1 present in chromanol ring of α-tocopherol (α-T) was replaced with a methylene group. This modification changed the electronic effects in the chroman-6-ol system (important for antioxidant function), which has recently been recognized quantitatively [8]. The obtained results of calculating the concentration required causing 50% of maximum effect (EC_50_), the antioxidant activity toward peroxyl radicals (ROO·), and the ability of the studied compounds to reduce the 2,2-diphenyl-1-picrylhydrazyl (DPPH) free radical using EPR spectroscopy confirmed that 1-carba analogues are much less effective scavenging agents than the parent chroman-6-ols. The newly synthesized carba analogue of α-T, which is devoid of the heterocyclic oxygen atom O1, may not only influence its antioxidant properties but also interfere with the intermolecular interactions with fatty-acid chains due to it lacking two nonbonding *p* orbitals, leading to changes in conformational dynamics and the structure of the membrane.

Numerous studies have shown the strong influence of α-T on the phase behavior of phospholipid bilayers in membranes [9,10,11]. It is common knowledge that those changes are mainly related to the type of fatty-acid chains, their saturation, and their length. However, it has been confirmed that the head groups and attached moieties also influence the membrane phase behavior [1]. Thus, taking into account already presented results [8] and possible interactions of modified chromanol ring with fatty-acid chains, we applied fluorescent probes embedded in a 1,2-dipalmitoyl-*sn*-glycero-3-phosphocholine (DPPC) liposome as reporters of changes in the biophysical properties of the membrane [12,13,14].

Temperature studies of liposomes and membranes with embedded cholesterols or drugs have shown that the measured fluorescence properties of environmentally sensitive hydrophobic dyes, providing estimates of fluorophore location, intensity, and polarization in the membrane, reflect the changing properties of membranes in terms of chain order, fluidity, hydration, or phase transitions [15].

Fluorescent dyes due to their various chemical structures and physicochemical properties are widely used to measure the physical properties of membranes [16]. Fluorophores are located in different regions of the membrane, revealing information regarding chain order, hydration, fluidity, electric potential of the inner part or surface of the membrane, dynamics, and phase transitions [14,17]. Fluorescence anisotropy studies with 1,6-diphenyl-1,3,5-hexatriene (DPH) were applied to determine the fluidity of the bilayer [18,19]. The fluorescent properties of 1-anilino-8-naphthalene sulfonate (ANS), such as the blue shift of fluorescence, increasing intensity, and fluorescence lifetime, are related to dye interaction in the hydrophobic environment of proteins or lipids [13]. ANS and DPH are used as reporters of the changing organization and dynamics of the bilayer induced during the interaction of two- or three-component systems of liposomes or membranes with embedded ligands [15,20,21].

To determine and understand the mechanism of the interactions between newly synthesized CT and phospholipids, we applied DPPC liposomes as a simple model of biological membranes to evaluate the influence of CT on membrane biophysical properties. The analysis was carried out using steady-state emission spectroscopy of ANS and fluorescence anisotropy of DPH to monitor hydration, dynamics, and bilayer order. Both fluorescent dyes have different binding depths and locations in the liposome. The thermodynamic parameters of the investigated samples were measured using the differential scanning calorimetry (DSC) method.

## 2. Results

### 2.1. ANS Fluorescence

#### 2.1.1. ANS Fluorescence Intensity

ANS emission is affected by solvent polarity, the viscosity of the medium, and electrostatic ion pair formation [22]. The latter process increases the emission intensity, with a blue shift of the emission maximum when the negatively charged sulfonate group of ANS binds nonspecifically to the cationic site of the headgroup molecules, which, in the case of DPPC, is a positively charged choline group. In less polar solvents, the intermolecular electron transfer lowers the intensity and shifts the maximum to lower energies. In aqueous solutions, another efficient radiationless process occurs, leading to very low quantum yield [23]. In nonaqueous polar solvents, the intensity increases with increasing hydrophobicity. NMR and X-ray diffraction studies have shown that, in lipid layers, ANS is localized in the vicinity of the lipid glycerol backbone, where the center of the aromatic ring penetrates at least to the level of the chain’s initial carbonyl group [24,25].

Figure 1a presents the temperature dependences of ANS fluorescence intensity at the emission maxima (FI_max_) during heating of the sample with the increasing presence of CT embedded in the DPPC membrane.

During heating in pure DPPC, the ANS emission varied depending on temperature. The lowest FI_max_ was observed in the gel phase below 28 °C, whereas a sharp increase in the intensity of FI_max_ was observed with a shoulder at 37 °C, reaching a maximum at around 42 °C. Further temperature increases caused a slow decrease in FI_max_. The increasing presence of CT in the bilayer increased FI_max_ in the gel phase, widened the trace, and flattened the curves with simultaneous loosening of its structure, including disappearance of the shoulder at 37 °C. The positions of FI_max_ maxima remained relatively constant, except at 20 mol%, where the shape changed significantly, indicating changes in the liposome structure. The maximum at 42 °C was ascribed to the main phase transition from the gel to the liquid phase, where FI_max_ decreased with increasing temperature, with similar slopes for low CT concentration traces. The traces of 20 mol% CT showed different behavior, suggesting substantial changes in structural organization in the liquid crystalline phase of DPPC. The observed FI_max_ quenching in this phase was ascribed to the increasing influx of water into this region. Such an explanation is supported by the fact that ANS emission was efficiently quenched by water molecules and, in the liquid crystalline phase, it was distorted by the presence of a high CT concentration; thus, a significant change in membrane ordering and increasing surface areas per lipid occurred, allowing water molecules to enter this area.

Figure 1b shows the relationship between the concentration of CT present in DPPC and the transition temperature (T_m_) or the slope obtained during fitting to Equation (1). The T_m_ obtained for pure DPPC from the ANS sensor, located in the interphase region, was 41.2 °C, which is similar to that reported for other DSC measurements [26,27]. This plot shows that decreases in T_m_ were linearly related to the CT concentration, which indicates that changes in the interface region, as detected by ANS fluorescence, also participated in the main phase transition. The value of the slope is mathematically related to the width of the transition that is correlated to the transition cooperativity, which is used as a measure of bilayer order [28]. The plot of the slope shows a sudden decrease at 2 and 5 mol% and minor changes at higher CT concentrations. Such behavior reflects the structural changes connected with increasing bilayer disorder. An indication of such changes is clearly visible in Figure 1a for 10 and 20 mol% traces, which have significantly different shapes compared to those of pure DPPC or 2 and 5 mol% CT.

#### 2.1.2. Position of ANS Emission Maxima (λ_max_) and Kinetics

During our temperature experiments, it appeared that considerable changes in the positions of ANS emission maxima (λ_max_) with increasing CT concentrations were observed. To estimate the relationship between λ_max_ and the microenvironment, we obtained calibration measurements of ANS emission in solvents with different protic and hydrophobic properties, from water to hexane (data not shown). It appeared that microenvironment changes of ANS from hydrophobic to aqueous were connected with the shift of λ_max_ to longer wavelengths. Figure 2a presents the changes in the positions of λ_max_ acquired during heating of the sample. It shows that, for pure DPPC in the gel phase, λ_max_ was located at 477 nm, indicating that emission originated from ANS molecules residing in an aqueous environment. Increasing temperature caused λ_max_ to shift to shorter wavelengths, reaching the lowest value at 42 °C, indicating that ANS molecules experienced a hydrophobic environment due to a shift into the membrane interior induced by the changing location of the choline group. Increasing temperature leading to phase transition shifted λ_max_ back to longer wavelengths, indicating that the ANS microenvironment became more aqueous. Increasing access of water molecules into the membrane interior occurred very probably due to loosening of the membrane structure in the liquid crystalline phase. We may also observe that the shape of the temperature profile mimics that obtained from ANS fluorescence intensity studies (see Figure 1a), showing a maximum at 42 °C and a pronounced shoulder at 35 °C. These temperatures are characteristic of main phase transition and pretransition in DPPC liposomes, as measured by the DSC method [28].

In the gel phase, increasing CT concentrations shifted λ_max_ to lower wavelengths, indicating that ANS molecules progressively populated the deeper, more hydrophobic interior of the bilayer. The lowest wavelength was measured at a CT concentration of 10 mol%, whereas λ_max_ increased at a CT concentration of 20 mol%, indicating that ANS molecules were located in the less hydrophobic environment. This is clear evidence that the 20 mol% CT concentration induced significant structural changes in the DPPC bilayer.

For the 10 and 20 mol% CT concentrations, there was no structure to the traces, which monotonically increased during the entire range of applied temperatures. However, one can observe that both CT traces exhibited a type of two-state phase transition in the modified membrane structure at around 32 °C. Such behavior indicates that the presence of more than 10 mol% CT in DPPC led to the formation different or modified bilayer structures. The plots also show that the new structures exhibited much better water inflow into the bilayer interior than pure DPPC liposomes.

Figure 2b shows temporal traces of ANS transfer into the DPPC membrane with increasing CT concentration. In pure DPPC, the process of binding was very fast, which was ascribed to parallel adsorption of ANS on the membrane surface in bulk water environment. With increasing CT concentration, the kinetic traces became two-component. The faster component was ascribed to the interactions between the membrane surface and ANS. The much slower component was ascribed to the diffusion of ANS molecules into the bilayer interior. The equilibrium observed at longer times was related to the accessible number of binding places and depended on the CT concentration. The fitting function for pure DPPC was monoexponential, whereas those with the presence of CT were double-exponential. The short-component lifetime was in the range of one-tenth of a second, whereas that of the longer component was in the range of 100 to a few hundred seconds. One may also notice that the shapes of the recorded kinetics were different, reflecting the different structural changes formed with different CT concentrations in DPPC. The kinetic measurements of ANS transport also confirmed the destructive effect of CT on DPPC membrane organization. The biphasic characteristic of ANS binding to liposomes has also been reported [27,29].

### 2.2. DPH Fluorescence Polarization

DPH is a fluorescent probe used to determine structural and dynamic properties of lipid bilayers. Emission anisotropy of DPH is interpreted as an indicator of lipid order or of the viscosity of the bilayer interior of the acyl chain and is also called fluidity [30,31,32,33]. In this study, DPH fluorescence anisotropy was used to determine DPPC membrane fluidity in the presence of CT.

Figure 3a displays the effects of temperature changes on the fluorescence anisotropy of DPH with increasing concentration of CT in the DPPC liposome. The obtained data were fitted with the Boltzmann function (Equation (1)), and calculated parameters of T_m_ and the slope are plotted in Figure 3c. The plot of pure DPPC shows a sigmoidal shape with an abrupt change between 40 °C and 43 °C and the T_m_ of DPPC at 41.4 °C. At increasing CT concentrations, the plots shifted to lower temperatures and flattened. Simultaneously, the onset of transitions in the gel phase shifted to lower temperatures. For up to 5 mol% CT, a consistent phase transition profile was observed. There were noticeable changes in the shape and slope of the traces recorded with 10 and 20 mol% CT. In particular, the trace for 20 mol% lost its sigmoidal shape, and the anisotropy decreased almost linearly with increasing temperature. Figure 3a also shows that, in the gel phase, the amount of incorporated CT did not significantly influence anisotropy, except at the 20 mol% concentration. At temperatures above T_m_ with increasing CT concentrations, the anisotropy values increased compared to pure DPPC, indicating increased viscosity in the formed structures.

The obtained temperature profile of anisotropy reflected the changes in the molecular order of packing of aliphatic alkyl chains induced by the presence of CT embedded in the DPPC bilayer. In the gel phase, the DPH molecule was located in the hydrophobic environment of lipid acyl tails, and its motion was restricted, leading to high anisotropy values. With increasing temperature, the rotational motion of DPH increased, leading to decreased anisotropy.

Figure 3b shows changes in DPH anisotropy in the gel phase (25 °C) and liquid crystalline phase (48 °C). In the gel phase, at increasing CT concentrations, there was very small progressive anisotropy lowering (0.01), indicating that membrane fluidity remained almost constant. This may indicate that embedded CT with its hydrophobic phytyl chain does not interrupt the ordered, rigid structure of the membrane phospholipids. In the liquid crystalline phase, with increasing CT concentration, an increase in the anisotropy amplitude of 0.05 compared to pure DPPC was observed. Thus, the presence of embedded CT in DPPC membranes in the LC phase decreased its fluidity. A similar effect of increasing anisotropy in the LC phase of DPPC was observed for α-T [34,35] and cholesterol [36,37].

The effect of increasing CT concentrations on slope values of fitted traces is presented on the left axis of Figure 3c, showing that this parameter linearly decreased with increasing CT up to 10 mol%. The first derivative of fitting function showed that the width of the band is inversely related to the slope of the original trace. Increasing the width of the band, i.e., lowering the value of the slope parameter, results in rising disorder in the bilayer and decreasing cooperativity during phase transitions. Thus, this suggests that inserted CT molecules induced disorder in the packing of alkyl chains, making the structure less rigid and, finally, lowering membrane fluidity.

The other plot in Figure 3c (right axis) presents the dependence of T_m_ versus CT concentration, showing that increasing the concentration of CT lowered the temperature of phase transition starting from 41.4 °C for pure DPPC. The temperature decrease was almost linear in the whole concentration range; however, at 10 mol% concentration, a parallel shift of this line was observed at 38.5 °C. Such a phenomenon indicates that a new structure was formed.

### 2.3. DSC Measurements

Differential scanning calorimetry is a widely used technique to accurately determine thermodynamic parameters during aggregation, phase transitions, or structural changes in proteins, biopolymers, and membranes. Liposomes undergo phase transition at specific temperatures T_m_, which strongly depends on the lipid composition forming the liposome. The presence of other ligands in the membrane introduces interactions leading to changes in its thermodynamic properties [38,39]. Therefore, we used this method to study interactions of CT, a tocopherol derivative, with a DPPC membrane.

#### 2.3.1. Thermodynamic Parameters

Figure 4a presents DSC scans of pure DPPC (2 mg/mL) and with 2, 5, and 10 mol% concentrations of CT. Pure DPPC exhibited two characteristic peaks at 36 °C and 42 °C, ascribed to pretransition and main phase transition, respectively. At 2 mol%, we can observe decreasing intensity and widening of the main peak, as well as the formation of a shoulder with a maximum at 40 °C. Presence of 5 mol% CT led to further changes in DSC shape. The intensity of the main peak decreased, and a new peak with a maximum at 39.5 °C appeared. The presence of 10 mol% CT led to 10 times lower intensity, a shift of the main maximum to 40 °C, its widening, and buildup of another peak with a maximum at 38 °C. This shows that the increasing presence of CT in DPPC caused a progressive decrease in intensity, increasing the width, and shifting the main phase transition to lower temperatures. At all CT concentrations, a pretransition peak was not detected. Attempts to record a DSC trace in the presence of 20 mol% CT failed; thus, it was not possible to register any phase transition in such a mixture.

Thermodynamic parameters which characterize interactions between CT and DPPC, such as T_m_, width of the main peak, and enthalpy of transition (∆H_m_), were obtained from DSC measurements. The results are presented in Figure 4b, showing that, at low concentration, T_m_ changed slowly, and a significant shift toward lower temperatures can be observed at 10 mol% CT. The order–disorder behavior of acyl chains in the bilayer is related to T_m_; thus, its decreasing value, especially at higher CT concentrations, indicates that its presence led to disruption of the membrane structure as a result of diminishing van der Waals interactions between acyl chains. We may also assume the contribution arising from polar interactions between phospholipid head groups and chromanol ring of CT devoid of heterocyclic oxygen. 

The width of the main peak is related to the cooperativity of the phase transition; accordingly, cooperativity was highest in pure DPPC, whereas the increasing presence of CT significantly decreased the cooperativity. For DPPC, it was assumed that the main transition from the gel phase to the liquid phase is a two-state process; thus, the transition was sharp and occurred with high cooperativity. Incorporation of CT into DPPC significantly increased the width of the main peak, indicating lowered cooperativity during the phase transition. The increasing presence of CT changed the shapes of the DSC scans, indicating their multicomponent structure connected with the formation of intermediate states or domains.

The results of the enthalpy calculations (Figure 4b) show a linear decrease with increasing CT concentration from 42 to 6 kJ/mol. The calculated ∆H_m_ associated with the melting of acyl chains in DPPC decreased with increasing CT concentration. Enthalpy is mainly related to melting of the acyl chain during transformation rearrangement from *trans* to *gauche* conformation; therefore, such progressive lowering of enthalpy indicates a decreasing number of *trans* molecules, i.e., the gel state, and an increasing number of *gauche* conformers, forming a liquid phase. Such a behavior indicates membrane melting due to the formation of liquid phases at lower temperatures.

The simultaneous decrease in intensity, widening of the peak, lowering of T_m_, and decrease in ∆H_m_ indicate increasing disorder in the membrane. Broadening of DSC peaks is associated with disrupting the membrane structure, increasing packing disorder, and significantly lowering the cooperativity of the phase transition. Thus, such symptoms confirmed that the increasing presence of CT led to membrane fluidization at lower temperatures compared to pure DPPC.

#### 2.3.2. Fitting of DSC Traces into Individual Components

The investigated system was complex; thus, to retrieve realistic data from measured thermograms, it was necessary to connect them with a known physical model of conformational transitions. In this approach, two-state and multistate transition mechanisms were used, and the applied model allowed for a detailed characterization of the obtained thermograms, providing descriptive parameters based on accepted thermally induced conformational transitions in membranes. Nevertheless, the deconvolution of thermograms into components allowed for qualitative determination of the mechanism occurring during melting of the membrane structure with increasing CT concentrations.

Figure 4a shows the DSC traces and their deconvolution into individual transitions obtained using the phenomenological model. The goal in this method was to use a combination of the appropriate number of peak logistic curves to reproduce the experimental curve. The traces were simulated assuming the two-state model for each fitted component. The height of the peak is related to the concentration and enthalpy of this transition. The fitting results show that increasing concentrations of CT widened the main peak and formed other phases with a simultaneous shift to lower temperatures of T_m_. We may observe that the gel–crystalline phase transition was present even at 5 mol% CT, with significantly decreasing intensity. At 10 mol% CT, this transition disappeared from the DSC trace, indicating fluidization of the membrane and the formation of other structures. We may also notice that, with the increasing presence of CT, the coexistence of different structures could be observed. Widening of the main phase transition peak observed at 2 mol% reflected the decreasing cooperativity of the transition due to the presence of CT. At 5 mol% CT embedded in the DPPC bilayer, another component with a maximum at 39.5 °C was observed, suggesting the formation of another phase or structure. The addition of greater than 10 mol% CT led to the appearance of another structure with a maximum at 38 °C. As shown, CT has a preference to form mixed structures with phase transition temperatures lower than that observed for pure DPPC.

During fitting procedures of the obtained DSC traces, we found different maxima ascribed to newly formed structures. Using this phenomenological approach, we could not assign those peaks to specific structures; however, taking into account the obtained results with possible interactions of components, we could ascribe them to the formation of CT-enriched domains, CT–phospholipid domains, and the formation of CT micelles.

### 2.4. Zeta Potential (ZP) Measurements

The value and the sign of the zeta potential (ZP) are measures of the net charge on the membrane surface; in the case of DPPC with its zwitterionic head, the sign and the magnitude of ZP depend on the ionic strength [40]. Thus, in buffer solutions at low ionic strength, the liposomes presented a small negative potential due to the exposed phosphate group, whereas, under a high ionic strength, the ZP was slightly positive.

Figure 5 shows the ZP of the DPPC surface with increasing concentrations of CT. In the given experimental conditions, pure DPPC exhibited a slightly negative potential of −4.2 mV, with similar values reported by others [41,42,43]. The addition of 2 mol% CT decreased ZP to a value of −2.0 mV, and its value did not change significantly with further increasing concentrations of CT, showing a slight decrease to −2 mV at 20 mol% CT. The shape of the concentration dependence and its values are comparable to those observed for tocopherol embedded in the DPPC bilayer, as reported in previous studies [42]. Surface charge properties are determined by the hydration shell. This effect can be observed for zwitterionic lipids where their net charge is equal to zero. Both moieties in the head group possess lone-pair electrons, thereby polarizing the water around them. Depending on the orientation of water molecules, they can modulate the orientation of the phosphate group. The observed small increase in ZP indicated the rearrangement of head-group moieties in the interface area, induced by the presence of CT [44,45].

### 2.5. Antioxidant Properties of CT

The performed measurements with the DPPH method confirmed the antioxidant properties of the carba analogues of α-T. The antiradical properties that allow for scavenging of the DPPH radicals of CT are presented in Figure 6. The results obtained for CT were compared with the original α-T sample. For the CT sample, after 15 min of incubation, the absorbance value decreased compared to the control sample, but its activity was much less effective than that observed for the original α-T sample. A similar observation for the carba analogues of Trolox was reported in our previous studies [8]. For both α-T and CT, longer incubation did not significantly affect the absorbance value.

## 3. Discussion

Application of calorimetric and spectroscopic techniques revealed that the presence of CT in a DPPC membrane changed its thermodynamic and structural parameters, which were measured as a function of temperature and concentration. Applied fluorescent probes bind at different positions in the membrane structure; thus, they report the modifications in the lipid microenvironment, reflecting changes induced by the presence of CT. To explore and interpret the results obtained from the fluorescence studies, in the framework of membrane phase transitions, the data were transformed into contour plots (Figure 7), where points with the same emission intensity were merged. The fast changes in colors and the contour lines indicate changes in the given parameter, signaling transitions or the formation of new phases. In all plots in Figure 7, along the ordinate at 0% CT we may observe three regions: (1) below 32 °C, ascribed to the gel phase, L_β_; (2) from 35 °C to 42 °C, ascribed to the ripple phase, P_β’_; (3) above 42 °C, ascribed to the fluid phase, L_α_. In Figure 7a, the maximum at 42 °C is anticipated as the main phase transition. The values are compatible with the data obtained from the DSC measurements (see Figure 4a).

The increasing ANS fluoresce intensity with the increasing presence of CT was related to the increasing number of binding sites in the interface membrane region, where loosening of the membrane structure due to inclusion of CT molecules provided access to more binding sites, whereas quenching of ANS FI_max_ was related to the increasing penetration of water molecules into the bilayer interior. The information from these data was used as a measure of packing density, which is understood as the distance between lipid molecules in the bilayer (Figure 7a). As shown in Figure 7a, the contour plot indicates a complex relationship between temperature and CT concentrations, indicating features not observed directly from Figure 1a. Above the transition temperature and at low CT concentrations, its presence decreased membrane packing density. This shows that fluorescence intensity was mostly affected at lower temperatures within the gel phase range, with the obvious formation of a new structure at 10 mol% CT. Interestingly, at higher CT concentrations and temperatures, the packing density increased, suggesting the formation of the ordered liquid phase. Similar results in the DPPC membrane with increasing cholesterol concentration were reported by Wang et al. [46].

The λ_max_ of ANS was related to the hydrophobicity of the microenvironment in the interphase/glycerol region (Figure 7b). The observed red shift of emission indicates decreasing hydrophobicity. Due to the strong sensitivity of ANS to water molecules, it served as an indicator of the presence of water molecules inside the bilayer interphase region. The hydrophobicity contours show that, in the gel phase, increasing CT concentration shifted ANS molecules into a more hydrophobic environment, revealing the formation of a new phase at 10 mol% CT. Above 35 °C, with increasing CT concentration, the ANS microenvironment became progressively less hydrophobic, indicating an influx of water molecules into the interface region, leading to fluidization of the bilayer with increasing temperature and concentration. This shows that the degree of hydration increased very sharply with increasing CT concentration. The significant role of water in the acyl chain region of liposomes has been confirmed by others [44,47].

Figure 7c shows the contour plot of DPH fluorescence anisotropy, which may be considered a representation of the membrane order–disorder transition described as conformational order in the acyl chain region, since the DPH molecule was located inside the acyl chain region where the main phase transition took place. It shows that increasing the presence of CT linearly decreased the temperature of the membrane order–disorder transition. We may notice that, at low CT concentrations, the transition temperature range between ordered and disordered phases was narrow, whereas, at higher CT concentrations, this range was much wider. The manifestation of nearly parallel lines to the abscissa above 35 °C in mixed and liquid phases indicates that the influence of the CT concentration on the degree of disorder was negligible, whereas it depended on increasing temperature.

The contour plots presented above indicate that embedded CT caused membrane disorder, leading to its fluidization, which increased with temperature and concentration. Similar results were obtained from the DSC measurements, as shown in Figure 4a, where the presence of CT decreased the main phase transition temperature, and the formation of new phases occurred. Additionally, we may notice that contour plots of DPH fluorescence anisotropy show that this parameter was a very good indicator of membrane disordering and fluidization processes; however, it could not detect any phase transitions. Phase transitions were readily detected from the FI_max_ contour plot of ANS since this molecule distinguished between binding to a hydrophobic environment and the bulk solution. The assignment of phase transitions from the λ_max_ contour plot was not that obvious; however, in this case, ANS signaled the presence of water molecules in its microenvironment. The ANS molecule was located inside the interphase region; thus, it sensed phase transition changes in this region. The decreasing T_m_ obtained from the ANS data indicates that the structural changes related to the melting process in the interphase region contributed during the main phase transition [15].

The presence of water molecules associated with lipids keeps the membrane in equilibrium and forms a dipole potential, which prevents fusion or adhesion of the membrane. The hydration process in the equilibrated membrane includes interaction with phosphate and carbonyl groups. The total or partial exclusion of water molecules or the reorganization of water molecules from that region leads to decreasing potential, thus allowing changes in bilayer geometry, leading to membrane structural changes. The results from the ZP measurements indicate that the presence of CT reduced the polarity of the phospholipid bilayer. Similar results were shown for the presence of cholesterol in phospholipid bilayers, and we showed that it decreases the relaxation rate of water molecules at the hydrophobic–hydrophilic bilayer interface [48].

The data from Figure 7a–c indicate that, at low CT concentrations and above T_m_, the mixture occurred in the most disordered state. At higher CT concentrations, the mixture became more fluid with increased ordering. At temperatures below 30 °C, the mixture was in the ordered gel phase, which decreased with increasing CT concentrations. This behavior is similar to that observed in phospholipid bilayers with the increasing presence of cholesterol and the formation of ordered liquid and disordered liquid phases and their mixtures [48,49]. In the temperature range of 35 °C to 50 °C, we may identify the region of two-phase coexistence related to the transition from ordered to disordered phases, which evolved with increasing temperature and CT concentrations. Moreover, DSC traces of CT:DPPC mixtures consisted of overlapped peaks (see Figure 4a), implying the coexistence of different phases. Collected views of contour plots (see Figure 7), showing continuous modification of measured parameters, confirmed the structural changes in the bilayer induced by CT. These changes were pronounced at higher CT concentrations, clearly indicating modification in the gel phase.

It is common knowledge that the presence of ligands in a bilayer structure leads to changes in the temperature of the main phase transition T_m_. Changes in this temperature were observed for ligands, which introduced disorder into the bilayer as a result of interactions in the acyl chain and the head group regions, leading to decreased packing density. The presented results show that increasing the CT concentration significantly decreased the T_m_ temperature. Such a behavior may be ascribed to the fact that lacking the heterocyclic oxygen atom of chromanol ring made this structure more hydrophobic compared to tocopherol, which enabled the CT molecule to more deeply penetrate into the DPPC interior. This modified configuration gave the CT molecule higher mobility, thus introducing more disorder into the bilayer structure. Chain melting starts from a single seed, which, in our case, could have been a CT molecule forming microdomains in the bilayer structure. Due to its structural similarity to the tocopherol molecule, including the presence of a hydroxyl group at the 6-position, we may assume that, in the DPPC membrane, CT located a modified chromanol ring in the vicinity of the glycerol backbone, which is lower compared to that reported for α-T [1,50,51]. Thus, the phytyl chain was embedded deeper in the hydrophobic region. This inclusion in the bilayer disturbed the geometry in this region, leading to increased disorder, as measured by the DSC method and DPH fluorescence. Deeper inclusion of the CT molecule is possible due to fact that this vitamin E analogue, containing the 1,2,3,4-tetrahydronaphthalene skeleton in which the heterocyclic oxygen atom is substituted by “nonparticipating” carbon atom at position C-1, leads to flattening of the heterocyclic ring compared to the half-chair conformation of the dihydropyranyl ring in tocopherol. Finally, CT expelled [52] water molecules from this region. In the liquid crystalline phase, we observed that increasing presence of CT decreased membrane ordering and lowered packing density, leading to increased hydration. The observed increased packing density and bilayer ordering at higher CT concentration above T_m_ was ascribed to water molecules entering the acyl chain region, which stabilized this region, potentially contributing to the observed increased bilayer stability [47,53].

As stated above, changes in the main phase transition reflected the interactions between acyl chains of phospholipids and embedded ligand. Changes in the measured thermotropic parameters in the presence of tocopherols indicated interactions between embedded tocopherols and the acyl chains of DPPC.

In DSC measurements, the increasing presence of tocopherols always led to broadening of ΔT_1/2_, progressively decreasing T_m_ and diminishing ΔH_m_ and cooperativity, together with the simultaneously formed new bands, thus indicating that the structural changes in the liposomes were induced by the presence of tocopherols, leading to a fluidization process at lower temperatures. Our studies in DPPC have shown that the presence of α-T at low concentration affects the membrane’s physicochemical properties to a much lesser degree than observed for other derivatives as oxalate, succinate, or CT [35,54,55]. Significant structural and morphological changes were observed at concentrations above 2 mol% in contrast to tocopherol derivatives, where membrane disordering was observed even at 1 mol% concentration. Similar results were reported from X-ray diffraction where, at very low concentrations, α-T had a negligible effect on the thermotropic properties of DPPC [56]. DPH studies for investigated derivatives showed that membrane disordering follows increasing concentrations, whereas, in the case of α-T, the membrane structure eroded slowly and significant fluidization occurred at 20 mol%. Furthermore, in the case of α-T [35] and CT, we noticed increasing ordering in the liquid crystalline phase, whereas the ester derivatives showed the opposite effect, i.e., they introduced some disordering into the gel phase [35,54].

In pure DPPC, a pretransition was also observed, corresponding to the transition from the lamellar gel phase to a rippled gel phase, which was mainly related to the polar region of the bilayer. In Figure 1a and Figure 7a, this transition can be seen for pure DPPC, whereas, above 2 mol% CT, it disappeared; similar results were obtained during DSC scans (see Figure 4a). Behavior similar to that of CT was observed in other tocopherol derivatives as oxalate or succinate, where the disappearance of pretransition at concentrations as low as 2 mol% was observed. In the case of α-T, the disappearance of pretransition was observed above 2 mol% [26]. The observed lower sensitivity to α-T compared to its derivatives is related to the fact that the presence of the modified chromanol ring of CT induced interactions in the polar region of DPPC. The observed efficient suppression of this pretransition, suggesting that any interference into the chromanol ring of α-T changed its electronic distribution, led to different organization and dynamics of the interfacial water molecules, leading to a surface disordering effect.

Regarding potential applications of this CT analogue, it has been shown that its antioxidant properties are much less effective as a scavenging agent and much less effective as an antioxidant than its parent compounds [8]. Biological activity is related to transport across membranes, which depends on many factors, including fluidity or surface charge [52]. Since, as shown in this study, CT appeared as a very effective disruptor of membrane structure, it may be used as a drug due to its similar properties to other tocopherol derivatives, such as succinate or oxalate [35,54].

## 4. Materials and Methods

### 4.1. Reagents

1-Carba-α-tocopherol ((2,5,7,8-tetramethyl-2-(4,8,12-timethyltridecyl)-1,2,3,4-tetrahydronaphtalen-6-ol)) (CT) and dl-α-tocopherol (α-T) were obtained according to a recently published procedure [8]. 1,2-Dipalmitoyl-*sn*-glycero-3-phosphocholine (DPPC), 1-anilino-8-naphthalene sulfonate (ANS), 1,6-diphenyl-1,3,5-hexatriene (DPH), and the 1,1-diphenyl-2-picrylhydrazyl (DPPH) radical, as well as chloroform and methanol (both spectroscopic grade), were purchased from SIGMA Chemical Co. (St. Louis, MO, USA). Double-deionized water produced with a MicroPure Water System (TKA, Niederelbert, Germany) was used as a solvent. The chemical structures of the tocopherols are shown in Figure 8.

### 4.2. Preparation of Vesicles

For the liposome preparation, dry DPPC and the studied tocopherol analogue were dissolved in chloroform and mixed in the required proportions (final concentrations of CT were 0–20 mol%). Next, the solvents were removed under vacuum at 50 °C with a rotary evaporator. The formed DPPC dry film was hydrated with double-distilled deionized water (pH 5.3, conductivity of <60 nS/cm) and vortexed for 30 min at 50 °C. In the subsequent step, all samples were extruded repeatedly 11 times through a 100 nm pore polycarbonate filter using a LiposoFast Basic LF-1 extruder (Avestin, Mannheim, Germany). For DSC measurements, the resultant liposomal suspension (2 mg/mL final phospholipid concentration) was dispersed for 2 min by ultrasonication in an ultrasonic bath (with power of 100 W) and stored at 0–4 °C for at least 12 h before the measurements. For ANS and DPH fluorescence measurements, the final concentration of the lipids was 0.08 mg/mL, with a mean liposome size of 100 nm which did not change significantly in the presence of CT, as confirmed by dynamic light scattering (DLS) measurements (data not shown) using a Zetasizer Nano (Malvern Instruments, Worcestershire, UK) at 20 °C at an angle of 90°. The zeta potential (ZP) was measured using the same Zetasizer by Malvern Instruments (Malvern Instruments, Worcestershire, UK) at 20 °C.

During the measurements, the lipid-to-ANS and lipid-to-DPH ratios were in the range of 100:1 or less, which is substantially lower than the binding capacity of DPPC for ANS or DPH. The obtained results demonstrated that bound ANS introduced negligible changes to the measured physicochemical and thermotropic properties of DPPC.

### 4.3. Spectroscopic Measurements

The steady-state emission spectra were obtained using a Shimadzu RF 5001PC fluorimeter (Shimadzu Corp., Kyoto, Japan) with an excitation wavelength of 380 nm. All spectroscopic measurements were performed in a 1 × 1 cm quartz cuvette in the temperature range of 20–60 °C. The temperature of the sample in the fluorimeter was controlled using a temperature unit adapter. During heating of the samples in the abovementioned range, the emission spectra of ANS were measured every 2 °C. Steady-state fluorescence anisotropy of DPH in liposomes (excitation at 360 nm and emission at 430 nm) was performed using a PerkinElmer LS555 (Perkin Elmer Corp., Norwalk, CT, USA), according to the included anisotropy procedure. The measurements were undertaken in the temperature range of 25–50 °C, with 0.5 °C steps. Each anisotropy readout was taken from 5 s measurements, and at least three such readouts were determined to calculate the mean value and its error at each temperature.

### 4.4. DPPH Method

The antioxidant activity was determined using a method involving DPPH free radicals [57]. One milliliter of the DPPH methanol solution (300 μM) was mixed with 4 mL of a methanol solution of α-T or CT (25 μM). The mixture was then incubated for a 30 min at room temperature in the dark. During incubation, a reduction of part of the DPPH free radical occurred, and the absorbance at 517 nm just after mixing, after 15 min of incubation, and after 30 min of incubation was measured. The absorbance was measured against methanol as a blank using a Shimadzu UV 1202 spectrophotometer (Shimadzu, Kyoto, Japan).

### 4.5. Differential Scanning Calorimetry (DSC)

Differential scanning calorimetry was performed using a DSC 7 (Perkin Elmer Corp., Norwalk, CT, USA) equipped with an Intracooler II and Pyris Software 10.1. Details of the performance of these experiments were published previously [35]. The sample pan was placed in the calorimeter and isothermally held at 10 °C for 5 min, before heating to 60 °C with a scanning rate of 2 °C/min. A series of three consecutive scans of the same sample was performed to ensure scan-to-scan reproducibility. The parameters of the peak temperature (T_m_), enthalpy (∆H_m_, J/g), and the width of the main peak, correlated to the cooperativity of the phase transition, were determined from the DSC curve.

### 4.6. Data Analysis and Fitting Procedures

All plots, figures, calculations, contour plots, and fitting procedures, including statistics of the plotted data, were prepared using the Origin program (ver. 8.5, OriginLab Corp., Northampton, MA, USA). All experiments were repeated at least in triplicate for samples from different series.

The fluorescence parameters of ANS or the fluorescence anisotropy of DPH plotted as a function of temperature generated a sigmoidal shape dependence, which was fitted by a Boltzmann equation curve (Equation (1)).
(1)$$F\left(T\right)=F_{g}+\frac{F_{L}-F_{g}}{1+\exp\left(\frac{T_{m}-T}{a}\right)},$$
where T_m_ is the transition temperature, F_g_ and F_L_ are the upper and lower limit of fluorescence intensity, respectively, and a is the slope.

## 5. Conclusions

In the gel phase, each CT molecule embedded in the DPPC bilayer changed its hydration by expelling water molecules, creating a cluster with neighboring phospholipids. Increasing the CT concentration generated more such clusters, which eventually joined together with phospholipids, forming a new structure. This process of new structure generation was observed as the formation of composed traces of DSC, lowering the intensity, enthalpy, cooperativity, and temperature of the main phase transitions. On a molecular level, the DPH anisotropy data indicated that the presence of CT induced changes in the acyl chain region by decreasing the order in this region. ANS fluorescence data indicated changes in packing density due to the evacuation of water from the interface region in the gel phase, whereas, in the liquid phase, quenching of ANS fluorescence was due to the increased influx of water molecules into this region. We may summarize that changes in physical membrane properties indicated that interactions between CT and membrane phospholipids occurred in the fatty-acid chain region, as well as in the interface region, with a significant presence of mobile water molecules. Comparisons between α-T and its derivatives indicated that, very probably, changes in electronic distribution in the chromanol ring induced different organization and dynamics of the interfacial water molecules, leading to a surface disordering effect.

## Figures and Tables

**Figure 1 molecules-26-02851-f001:**
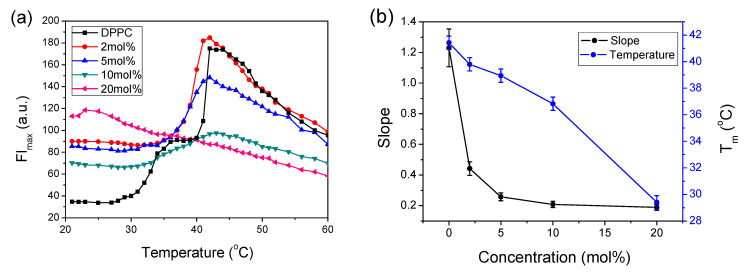
Changes in the ANS fluorescence intensity at the emission maximum (FI_max_) versus temperature with the increasing presence of CT in 0.08 mg/mL DPPC: (**a**) during heating from 20 °C to 60 °C; (**b**) plot of fitted parameters to Equation (1) of transition temperature (T_m_) and slope obtained during heating versus CT concentrations. ΔT = 0.5 °C.

**Figure 2 molecules-26-02851-f002:**
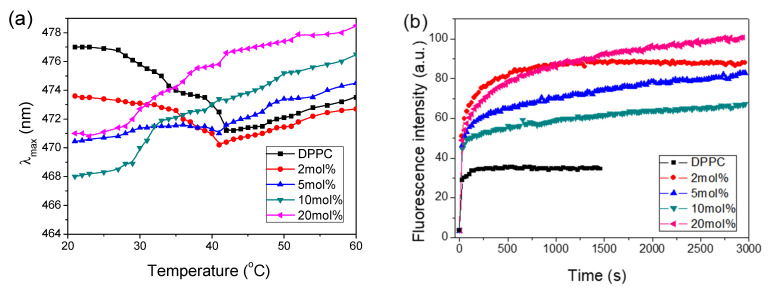
(**a**) Positions of ANS emission maxima (λ_max_) versus temperature with increasing presence of CT in 0.08 mg/mL DPPC during heating from 20 °C to 60 °C; (**b**) kinetics of ANS transport into the DPPC membrane at 25 °C. The concentration of CT is given in the legend.

**Figure 3 molecules-26-02851-f003:**
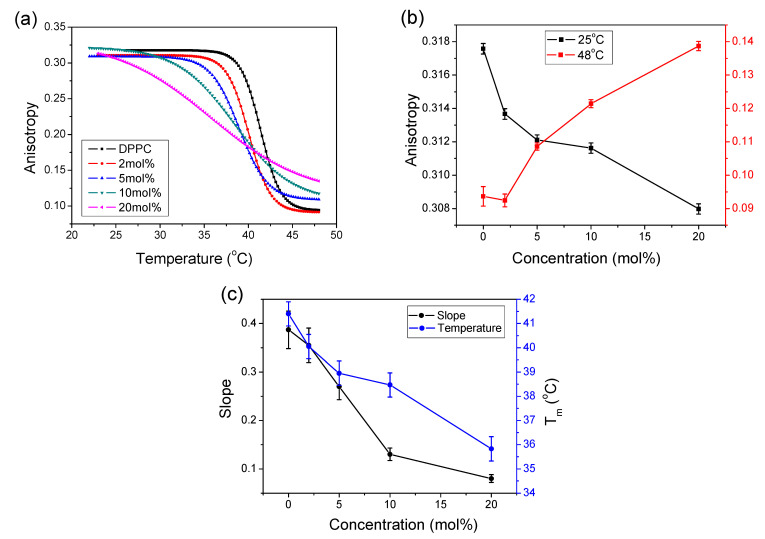
Effect of CT on DPH anisotropy in DPPC liposome: (**a**) temperature profiles of DPH anisotropy in 0.08 mg/mL DPPC at increasing concentrations of CT are given in the legend; (**b**) changes in DPH anisotropy in the gel phase at 25 °C and in the liquid crystalline phase at 48 °C; (**c**) plot of parameters fitted to Equation (1); T_m_, the temperature of the phase transition (right axis) and the slope (left axis). ΔT = 0.5 °C.

**Figure 4 molecules-26-02851-f004:**
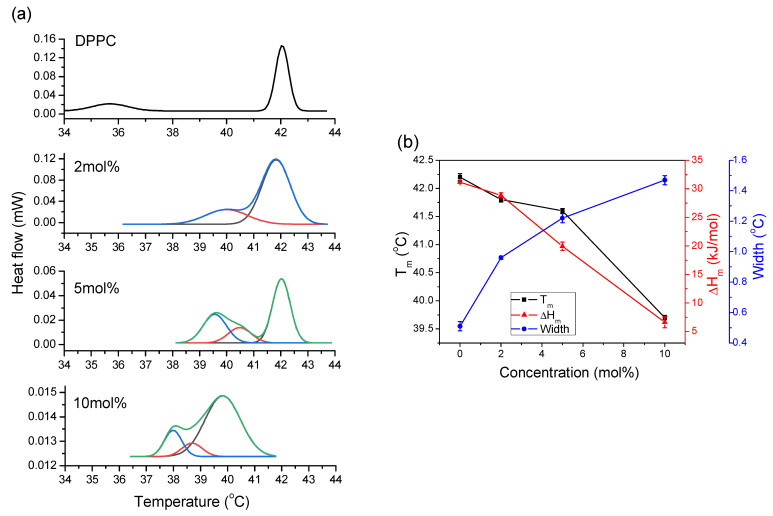
(**a**) Smoothed DSC traces of DPPC liposomes (2 mg/mL), with an increasing concentration of CT, given in the plot. The plot also shows the fitted components to the total DSC trace. Please note the decreasing intensity on the ordinate axis; (**b**) effect of incorporation of different concentrations of CT into DPPC on the main phase transition temperature, T_m_, the width of the main peak, and the enthalpy of transition, ∆H_m_. Please note decreasing values on *y*-axis plots.

**Figure 5 molecules-26-02851-f005:**
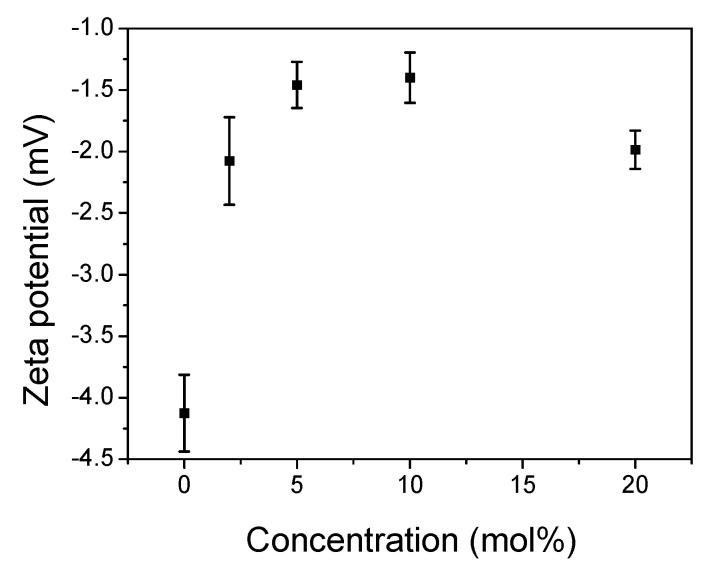
Zeta potential (ZP) of the 0.08 mg/mL DPPC membrane in the presence of increasing amounts of CT. Error bars are SDs calculated for three different repetitions.

**Figure 6 molecules-26-02851-f006:**
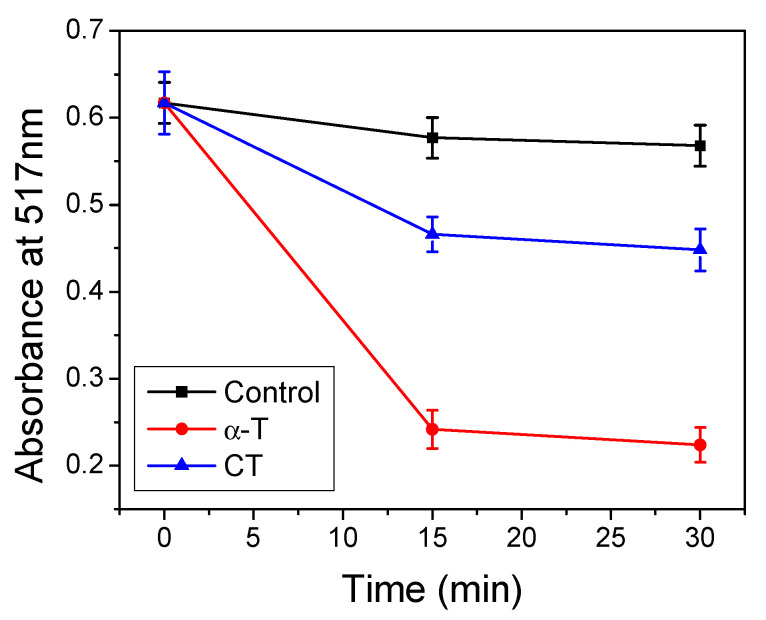
Absorbance at 517 nm of DPPH radicals in methanol in the presence of CT or α-T after 15 and 30 min of incubation. Indicated values represent the means ± SD arising from triplicate experiments.

**Figure 7 molecules-26-02851-f007:**
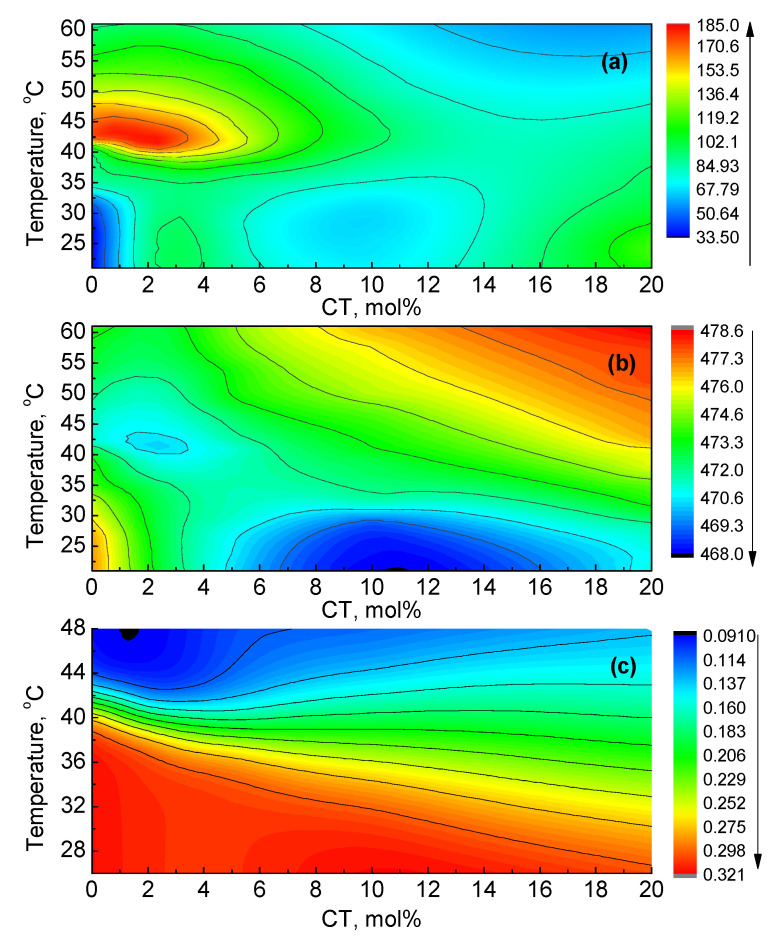
Contour plots of ANS and DPH; (**a**) packing density, using ANS FI_max_ data from Figure 1a; (**b**) hydrophobicity, using ANS λ_max_ data from Figure 2a; (**c**) order–disorder, using data from DPH emission anisotropy given in Figure 3a.

**Figure 8 molecules-26-02851-f008:**
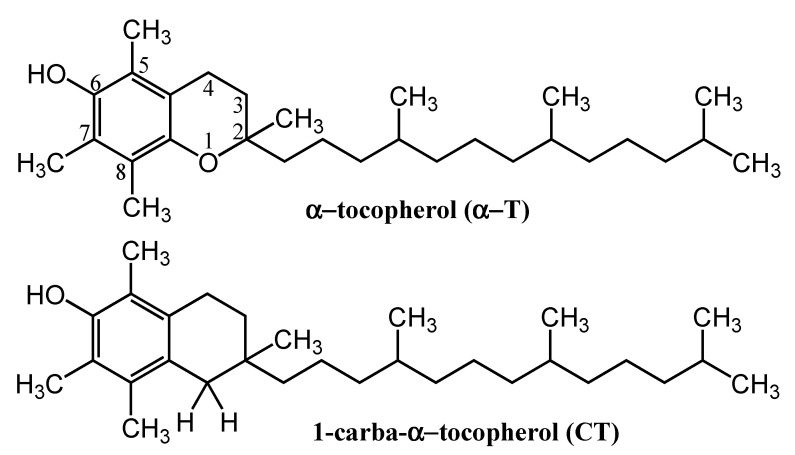
Chemical structures of α-tocopherol (α-T) and 1-carba-α-tocopherol (CT).

## Data Availability

The data presented in this study are available on request from the corresponding author.

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
