# Peer review of "Phase Transitions and Structural Changes in DPPC Liposomes Induced by a 1-Carba-Alpha-Tocopherol Analogue"

_molecules, 2021, doi:10.3390/molecules26102851_

Round 1

Reviewer 1 Report

Neunert and coworkers investigated phase transitions and structural changes in DPPC liposomes induced by a 1-carba-alpha-tocopherol analogue. Different characterization methods were used and the results are interesting. I recommend it for publication in Molecules after the following issues are addressed.

  1. Line 52, EC50 should be changed to EC50, and the full name of EC50 should appear first.
  2. Line 62-63, ‘1,2-Dipalmitoyl-62 sn-glycero-3-phosphocholine’ should be changed to ‘1,2-dipalmitoyl-62 sn-glycero-3-phosphocholine’.
  3. Line 354-368, the reason of the zeta potential of DPPC liposomes decreasing by adding CT should be added in the text.
  4. The resolution of the figures (figure 1-6) is not enough for publication.
  5. The references in this ms. are old. Some recent studies (doi.org/10.1016/j.colsurfb.2019.110422; doi.org/10.1021/acs.langmuir.0c00472) should be included.

Author Response

We have revised the manuscript accordingly, and detailed corrections are listed below point by point:

Response to Reviewer 1 Comments

Neunert and coworkers investigated phase transitions and structural changes in DPPC liposomes induced by a 1-carba-alpha-tocopherol analogue. Different characterization methods were used and the results are interesting. I recommend it for publication in Molecules after the following issues are addressed.

  1. Line 52, EC50 should be changed to EC50, and the full name of EC50 should appear first.

Response 1:

Lines 54-55 in new revised version of the manuscript

We made the requested supplement in the text in lines 54-55.

 “The obtained results of calculated the concentration required causing 50% of maximum effect (EC50) values..”

  1. Line 62-63, ‘1,2-Dipalmitoyl-62 sn-glycero-3-phosphocholine’ should be changed to ‘1,2-dipalmitoyl-62 sn-glycero-3-phosphocholine’.

Response 2:

Line 69 in new revised version of the manuscript

The requested correction was made in the text in line 69.

  1. Line 354-368, the reason of the zeta potential of DPPC liposomes decreasing by adding CT should be added in the text.

Response 3:

Lines 358-364 in new revised version of the manuscript

The section 2.4 was changed and some sentences was added:  

Similarity arises from the fact that part of both molecules, which influences the surface zeta potential, have an OH group at the 6 position. Surface charge properties are determined by hydration shell. The effect is observed for zwitterionic lipids where their net charge is equal to zero. Both moieties in head group possess lone pair electrons what polarizes the water around them. Depending on orientation of water molecules they can modulate orientation of phosphate group. Observed small increase in ZP indicates rearrangement of head group moieties in interface area induced by the presence of CT [44, 45].”

  1. The resolution of the figures (figure 1-6) is not enough for publication.

Response 4:

The resolution of the figures has been corrected in accordance with the publisher's guidelines (a resolution of 300 dpi or higher)

  1. The references in this ms. are old. Some recent studies (doi.org/10.1016/j.colsurfb.2019.110422; doi.org/10.1021/acs.langmuir.0c00472) should be included.

Response 5:

Some resent studies were added.

[14] Kafle, A.; Akamatsu, M.; Bhadani, A.; Sakai, K.; Kaise, C.; Kaneko, T.; Sakai, H. Effects of β-Sitosteryl Sulfate on the Properties of DPPC Liposomes. J. Oleo Sci 2018, 67, 1511–1519, doi:10.5650/jos.ess18147.

[38] Jovanović, A.A.; Balanč, B.D.; Djordjević, V.B.; Ota, A.; Skrt, M.; Šavikin, K.P.; Bugarski, B.M.; Nedović, V.A.; Ulrih, N.P. Effect of gentisic acid on the structural-functional properties of liposomes incorporating β-sitosterol. Colloids Surfaces B Biointerfaces 2019, 183, doi:10.1016/j.colsurfb.2019.110422.

[39] González-Ortega, R.; Šturm, L.; Skrt, M.; Di Mattia, C.D.; Pittia, P.; Poklar Ulrih, N. Liposomal Encapsulation of Oleuropein and an Olive Leaf Extract: Molecular Interactions, Antioxidant Effects and Applications in Model Food Systems. Food Biophys. 2021, 16, 84–97, doi:10.1007/s11483-020-09650-y.

We hope that introduced changes and given explanations fulfil reviewers expectations and that improved manuscript will be acceptable to be published.  

Sincerely,

Authors 

Reviewer 2 Report

In this manuscript the authors report on the effect of a tocopherol derivative on the phase behaviour of DPPC liposomes. The authors have previously characterized the effects of tocopherol and of several tocopherol derivatives (references 35 and 50 in the submitted manuscript), so have other authors (eg reference 34 in the submitted manuscript). The similarities and differences must be clearly indicated and discussed. The number of possible tocopherol derivatives is endless, and the relevance of this work is not clear.

Why is it important to study the effect of this derivative in particular on the phase behaviour of DPPC bilayers? What does this work adds to scientific knowledge?

In addition to this general concern, some specific concerns are indicated below by order of appearance in the manuscript:

1 – The writing should be revised. In particular the text found be more focused on the relevant message, avoiding unnecessary repetitions (eg lines 70‐72 in the introduction and most of the discussion) and clarifying the concepts (eg line 100, or “influx of water into this region” in lines 123‐124, “bonding sites” and information in lines 126‐129 (do the authors mean binding? Lipid bilayers do not have well defined binding sites), or “permeation into the membrane interior” (permeation is through the lipid bilayer, do the authors mean partition? Is ANS expected to locate in the interior of the lipid bilayer in spite of its negative charge?, among several other…).

Also, some typos should be corrected, (eg. “ion par” in line 95 or “traces of 20 mol%” in line121). The references should be verified, eg in reference 33 it is reported the effect of cholesterol,not of tocopherol as indicated in the submitted manuscript.

2 – Liposome preparation and characterization. The liposomes used have been obtained by ultrasonication of the hydrated lipids. The authors must provide the power used and duration of the ultrasonication process. It is also necessary to show the size distribution of the liposome suspension, both immediately after preparation and before measurement (after the 12 h incubation at 0‐4 C). In addition, the authors must justify why is the liposome suspension incubated at a temperature well below the lipid Tm. It is well known that lipid bilayers tend to exclude solutes when they are in very ordered phases. The authors must therefore show that the added tocopherol derivative remains in the lipid bilayer when the studies are performed.

3 – Possible formation of tocopherol aggregates. The authors must show that at the very high concentrations of tocopherol derivative used it is still solubilized in the DPPC bilayer and not as aggregates. This is of fundamental importance for the interpretation of the results shown, especially those reported by ANS and DSC.

4 – Results and interpretation of Figure 1. It is not clear how was the value of the Tm displayed on plot B obtained from plot A. It is also not clear to what temperature range refers the slope reported. The authors should give an example on this figure or as SI. Also, the cooling curves and the heating curve immediately after cooling must be shown. This is essential to show that the results obtained are not a consequence of incubation at low temperatures (see comments 2 and 3). As a minor detail, the y axis of plot b should be switched for easier reading of the figure (same in Figure 3 plot c).

5 – Results and interpretation of Figure 2. The kinetics of ANS fluorescence increase is much slower than the results previously reported for DMPC (reference 28 in the submitted manuscript). The authors must provide an interpretation for this observation. It is of particular importance to show that the slow kinetics and the higher extent of fluorescence increase is not due to the interaction of ANS with tocopherol aggregates (see comments above). It is also necessary to clearly indicate how long was the equilibration of ANS with the liposomes before measuring the data reported in plot a (as well as for all additional results with ANS). Were the liposomes previously equilibrated at the reported temperature or does the kinetics includes the lipid bilayer equilibration in addition to ANS interaction?

6 – DSC results. As indicated in comment 4, the cooling scan, consecutive heating scan must be shown. If changes are noticed, cooling and heating scans must be repeated until a reproducible result is obtained. The authors must also clearly identify the number of replicates performed and if they are readings from the same sample preparation or independent replicates. This
applies to DSC results and to all information reported in the manuscript.

7 – Zeta potential results. What is the relevance of discussing the effect of the ionic strength in the zeta potential of the lipid bilayers? Is the effect reported exclusive for zwitterionic lipids (as suggested)? Is the ionic strength originated from buffer different from that originated by other salts? Why does the OH group contributes to the zeta potential? The presentation and
interpretation of those results must be significantly improved.

8 – Discussion should be revised, it should be shortened and focused, a critical comparison with the results obtained for tocopherol and tocopherol derivatives should be done.

9 – Conclusion. The eventual formation of tocopherol aggregates must be unequivocally disregarded so are non‐equilibrium effects due to incubation of the DPPC SUVs at 0‐4 oC. Without this information the interpretation of the results obtained are speculation.

Given the comments above, major revisions are required before this work may be considered for publication.

Author Response

We have revised the manuscript accordingly, and detailed corrections are listed below point by point:

Response to Reviewer 2 Comments

In this manuscript the authors report on the effect of a tocopherol derivative on the phase behaviour of DPPC liposomes. The authors have previously characterized the effects of tocopherol and of several tocopherol derivatives (references 35 and 50 in the submitted manuscript), so have other authors (eg reference 34 in the submitted manuscript). The similarities and differences must be clearly indicated and discussed. The number of possible tocopherol derivatives is endless, and the relevance of this work is not clear.

To clarify relevance of this work in Introduction more explanation have been added (all changes are given by tracking changes in red colour).

Why is it important to study the effect of this derivative in particular on the phase behaviour of DPPC bilayers? What does this work adds to scientific knowledge?

More explanation has been added into introduction part, all changes are given in red colour.

In addition to this general concern, some specific concerns are indicated below by order of appearance in the manuscript:

1 – The writing should be revised. In particular the text found be more focused on the relevant message, avoiding unnecessary repetitions (eg lines 70‐72 in the introduction and most of the discussion) and clarifying the concepts (eg line 100, or “influx of water into this region” in lines 123‐124, “bonding sites” and information in lines 126‐129 (do the authors mean binding? Lipid bilayers do not have well defined binding sites), or “permeation into the membrane interior” (permeation is through the lipid bilayer, do the authors mean partition? Is ANS expected to locate in the interior of the lipid bilayer in spite of its negative charge?, among several other…).

Lines 70 -71 avoiding unnecessary repetitions

Response 1:

Lines 76-80 in new revised version of the manuscript

This fragment in text was changed to:

“Fluorescent dyes due to their various chemical structures and physico-chemical properties are widely used to measure the physical properties of membranes, such as surface membrane fluidity, hydrophobicity, polarity of the surface or inner membrane, and permeability [16]. Due to their various chemical structures and physico-chemical properties, Fluorophores are located in different regions of the membrane revealing information regarding chain order, hydration, fluidity, electric potential of the inner part or surface of the membrane, dynamics, and phase transitions [14,17]. “

Line 123-124   influx of water into this region

Lines 129-132 in new revised version of the manuscript

To this fragment it was added explanation in lines 129-132 in new revised version of the manuscript:

“Such explanation is supported by fact that ANS emission is efficiently quenched by water molecules and in liquid crystalline phase it is distorted by presence of high CT concentration thus significant change of membrane ordering and increasing surface areas per lipid occurs allowing water molecules enter this area. “

 “bonding sites” and information in lines 126‐129 (do the authors mean binding? Lipid bilayers do not have well defined binding sites).

As bonding sites for ANS molecules we consider positively charged choline head groups. It seems that “binding site” will be more appropriate expression.

“Permeation into the membrane interior” (permeation is through the lipid bilayer, do the authors mean partition?

Lines 159-160 in new revised version of the manuscript

Permeation has been replaced with more adequate word as partition or location. This fragment in text was changed to:

“ … indicating that ANS molecules experience a hydrophobic environment due to its shift  facilitated permeation into the membrane interior induced by changing position of choline group.”

Is ANS expected to locate in the interior of the lipid bilayer in spite of its negative charge?

Lines 100-104 in new revised version of the manuscript

ANS molecule in lipid bilayer is located in interphase region and its negative charge is on the level of choline head group, this has been described in lines 95 to 99 in old version of the manuscript and in lines 100-104 in its new revised version. Observed changes in fluorescence intensity are related to number of binding sites and presence of water molecules, whereas shifting of emission maxima is related to changing position of positively charged choline group in head group area due to temperature and/or CT concentration changes. This shifts ANS molecule  to more or less hydrophobic environment observed as changes of emission maximum. Emission transition moment of ANS is located inside naphthalene structure which is closer to glycerol and even in the vicinity of C1 acyl chain region. In this region dielectric permittivity changes occur sharply thus even small movement of ANS body results in noticeable shift of emission maxima. 

Also, some typos should be corrected, (eg. “ion par” in line 95 or “traces of 20 mol%” in line121).

Line 100 in new revised version of the manuscript

Corrected “ion pair”  ;

“traces of 20 mol%” I am sorry, but how  it should be corrected?

The references should be verified, eg. in reference 33 it is reported the effect of cholesterol,not of tocopherol as indicated in the submitted manuscript.

Line 240 in new revised version of the manuscript

A mistake was made in the text - an incorrect literature item was cited. We made the correction and the reference was transferred to “cholesterol effect”.

2 – Liposome preparation and characterization. The liposomes used have been obtained by ultrasonication of the hydrated lipids. The authors must provide the power used and duration of the ultrasonication process.

Response 2:

Lines 563-564 in new revised version of the manuscript

As stated in the line 563 of the manuscript: “…was dispersed for 2 min by ultrasonication in an ultrasonic bath (with power of 100 W)…” the sonication time was 2 min. The power of ultrasonication was added in the text in line 564 in new revised version of the manuscript.

It is also necessary to show the size distribution of the liposome suspension, both immediately after preparation and before measurement (after the 12 h incubation at 0‐4 °C). In addition, the authors must justify why is the liposome suspension incubated at a temperature well below the lipid Tm. It is well known that lipid bilayers tend to exclude solutes when they are in very ordered phases. The authors must therefore show that the added tocopherol derivative remains in the lipid bilayer when the studies are performed.

Based on our earlier studies and hydrophobicity nature of α-T and different α-T derivatives we have assumed that all amount of tocopherols has embedded into liposomes and the samples are stable with a few days. The method of preparation and storage gave reproducible phase behaviour when samples prepared at different times were examined also by calorimetry method and also no significant change in fluorescence properties or value of the hydrodynamic diameter of the obtained liposomes was observed. See also Response 3.

3 – Possible formation of tocopherol aggregates. The authors must show that at the very high concentrations of tocopherol derivative used it is still solubilized in the DPPC bilayer and not as aggregates. This is of fundamental importance for the interpretation of the results shown, especially those reported by ANS and DSC.

Response 3:

Measurements of spectroscopic properties for the tested concentrations were made in water and in liposomes. No measurable fluorescence emission was observed in water for any of the CT concentrations. Analogous measurements made in the liposome suspension showed a linear increase in fluorescence intensity with increasing CT concentration. As α-T is practically insoluble in water, the recorded fluorescence can be assumed to arise mainly from the monomeric form of α-T embedded into the DPPC bilayer. Moreover, the measured values of the CT lifetime in liposomes at different CT concentrations did not show significant differences in its values.

The dependencies of fluorescence emission maximum and lifetimes versus concentration of CT in the liposome suspension are given below.

4 – Results and interpretation of Figure 1. It is not clear how was the value of the Tm displayed on plot B obtained from plot A. It is also not clear to what temperature range refers the slope reported. The authors should give an example on this figure or as SI.

Response 4:

As stated in the lines 610-614 of the manuscript Tm and the slope was obtained by fitted by a Boltzmann equation curve (Equation (1)) to experimental curves from Figure 1A. Thus the data presented in Figure 1B are the values obtained from fitting procedures and we presented them in graphical form.  

Also, the cooling curves and the heating curve immediately after cooling must be shown. This is essential to show that the results obtained are not a consequence of incubation at low temperatures (see comments 2 and 3).

For all samples the heating and cooling cycles were repeat a few times. After cooling the next heating process has given similar course with limits of experimental error.

As a minor detail, the y axis of plot b should be switched for easier reading of the figure (same in Figure 3 plot c).

As reviewer suggested, we have switched y axis of  Figure 1b and Figure 3c.

5 – Results and interpretation of Figure 2. The kinetics of ANS fluorescence increase is much slower than the results previously reported for DMPC (reference 28 in the submitted manuscript). The authors must provide an interpretation for this observation. It is of particular importance to show that the slow kinetics and the higher extent of fluorescence increase is not due to the interaction of ANS with tocopherol aggregates (see comments above). It is also necessary to clearly indicate how long was the equilibration of ANS with the liposomes before measuring the data reported in plot a (as well as for all additional results with ANS).

Response 5:

The short value presented for DMPC arises from the applied method which allowed measurements with such millisecond resolution whereas during our measurements we may estimate time resolution in order of second. So, it is quite probable that short component obtained for DPPC is in this same range of milliseconds. The slow kinetics observed in gel phase at constant temperature reflects process of slow penetration of ANS into DPPC with similar rates whereas increasing intensity reflects rising number of electrostatic binding sites.

Regarding data for ANS measurements – the data for each point were scanned until no changes were observed.

As we have shown and confirmed by above presented fluorescence and DSC data given above no tocopherol aggregates were observed and no such structures were detected.

Were the liposomes previously equilibrated at the reported temperature or does the kinetics includes the lipid bilayer equilibration in addition to ANS interaction?

We always before any measurements equilibrate sample to measurement temperature, where possible.

6 – DSC results. As indicated in comment 4, the cooling scan, consecutive heating scan must be shown. If changes are noticed, cooling and heating scans must be repeated until a reproducible result is obtained. The authors must also clearly identify the number of replicates performed and if they are readings from the same sample preparation or independent replicates. This
applies to DSC results and to all information reported in the manuscript.

Response 6:

As stated in the line 608 of the manuscript: “All experiments were repeated at least in triplicate.” And the replicates was for samples from different series. 

In case of DSC measurements, as stated in the line 638 in old version of the manuscript:

“Three replicates were analyzed for each sample.”

For a better explanation, in lines 608-609 in the revision version of the manuscript, the following was added:

“All experiments were repeated at least in triplicate for samples from different series .”

And in line 600-601 we changed the sentence:

Three replicates were analyzed for each sample.” “A series of three consecutive scans of the same sample were performed to ensure scan-to-scan reproducibility.”

Heating and cooling scan as shown in attached Figure are very similar and when repeated in cycles heating – cooling – heating the obtained traces are reproducible within experimental error. These results are reproducible with this same samples or newly prepared samples regardless of rate of heating or cooling. The only observed differences between heating and cooling traces is temperature shift at maximum. On cooling an undercooling effect occurred with hysteresis of 2 oC and its manifestation in general is connected with presence of other molecules and in this case with presence of CT. Since the other thermodynamic parameters calculated from heating or cooling traces are very similar with limits of experimental error we plotted the results obtained during heating of the sample.

Raw DSC data for 2 mol% CT in DPPC obtained with heating-cooling-heating cycles; lower row – another sample

7 – Zeta potential results. What is the relevance of discussing the effect of the ionic strength in the zeta potential of the lipid bilayers? Is the effect reported exclusive for zwitterionic lipids (as suggested)? Is the ionic strength originated from buffer different from that originated by other salts? Why does the OH group contributes to the zeta potential? The presentation and
interpretation of those results must be significantly improved.

Response 7:

Lines 358-364 in new revised version of the manuscript

Sentence from line 358  was removed and more adequate explanation was added in lines 358-364: 

Similarity arises from the fact that part of both molecules, which influences the surface zeta potential, have an OH group at the 6 position. Surface charge properties are determined by hydration shell. The effect is observed for zwitterionic lipids where their net charge is equal to zero. Both moieties in head group possess lone pair electrons what polarizes the water around them. Depending on orientation of water molecules they can modulate orientation of phosphate group. Observed small increase in ZP indicates rearrangement of head group moieties in interface area induced by the presence of CT [44, 45].”

8 – Discussion should be revised, it should be shortened and focused, a critical comparison with the results obtained for tocopherol and tocopherol derivatives should be done.

Response 8:

Discussion has been revised including comparison between tocopherol and CT.

9 – Conclusion. The eventual formation of tocopherol aggregates must be unequivocally disregarded so are non‐equilibrium effects due to incubation of the DPPC SUVs at 0‐4 oC. Without this information the interpretation of the results obtained are speculation.

Response 9:

In our opinion we believe that presented results and answers and given explanations have unequivocally shown that presented interpretation of obtained results is sound and correct.

We hope that introduced changes and given explanations fulfil reviewers expectations and that improved manuscript will be acceptable to be published.  

Sincerely,

Authors

Reviewer 3 Report

The present manuscript deals with the study of the interaction of a novel tocopherol analogue by fluorescence spectroscopy and DSC on DPPC liposomes. In general, the manuscript is suitable for publication after some improvements.

Introduction: the authors should highlight better the relevance of studying the interaction between the new analogue and DPPC liposomes

Methods: Line 531-532 and 538-539 are results and they should moved to results section. Moreover, the authors should report the results on liposome size from DLS and indicate how a mean particle size of 100 nm was achieved (for instance sonication, extrusion...).

Why the liposome concentration is lower for fluorescence spectroscopy than DSC (0.08 mg/mL and 2 mg/mL, respctively)? Is related to some limitations about the technique used?

Line 562-563 The sentence is not understandable and reference [32] is about fluorescence spectroscopy and not DSC.

Discussion: the authors should detailed in the method section how the contour plots were built up and the software used. Moreover, the difference between alfa-tocopherol and CT analogue in the interaction with DPPC bilayer should be discussed in more details.

Author Response

We have revised the manuscript accordingly, and detailed corrections are listed below point by point:

Response to Reviewer 3 Comments

Reviewer 3:

The present manuscript deals with the study of the interaction of a novel tocopherol analogue by fluorescence spectroscopy and DSC on DPPC liposomes. In general, the manuscript is suitable for publication after some improvements.

Introduction: the authors should highlight better the relevance of studying the interaction between the new analogue and DPPC liposomes

To clarify relevance of this work more explanation has been added into introduction part, all changes are given in red.

Methods: Line 531-532 and 538-539 are results and they should moved to results section. Moreover, the authors should report the results on liposome size from DLS and indicate how a mean particle size of 100 nm was achieved (for instance sonication, extrusion...).

Response:

As stated in the line 608 of the manuscript: “All experiments were repeated at least in triplicate.” The mean values of liposome size 100 nm was determined from analysis of number of peaks from three different samples (from different series) and didn’t changed significantly in the present of CT (results also from three analysis).

The dependencies of mean liposome size versus CT concentration and example of number of peaks distribution for different concentration of CT for one series of samples are given below.

Because these results do not bring any significant changes in the data analysis, these results were not included in the manuscript and in the method section gives the mean value of the hydrodynamic diameter of the obtained liposomes (lines 566-567).

Lines 559-561 in new revised version of the manuscript

After preparation the samples were extruded through a 100 nm pore polycarbonate filter. This step of extrusion method was omitted in the part “Preparation of vesicles” what was corrected in the revision version of the manuscript.

We made the necessary additions in the text in lines 559-561:

The formed DPPC dry film was hydrated with double-distilled deionized water (pH 5.3, conductivity of <60 nS/cm) and vortexed for 30 min at 50 °C. In the following step, all samples were extruded repeatedly eleven times through 100 nm pore polycarbonate filter using LiposoFast Basic LF-1 extruder (Avestin, Mannheim, Germany). For DSC measurements, the resultant liposomal suspension (2 mg/ml final phospholipid concentration) was dispersed for 2 min by ultrasonication in an ultrasonic bath (with power of 100 W) and stored at 0–4 °C for at least 12 h before the measurements. For ANS and DPH fluorescence measurements, the final concentration of the lipids was 0.08 mg/ml, with a mean liposome size of 100 nm which didn’t changed significantly in the present of CT, as confirmed by dynamic light scattering (DLS) measurements (data not shown) using a Zetasizer Nano (Malvern Instruments, Worcestershire, UK) at 20 °C under an angle of 90°.

Why the liposome concentration is lower for fluorescence spectroscopy than DSC (0.08 mg/mL and 2 mg/mL, respctively)? Is related to some limitations about the technique used?

Response:

For DSC measurements the liposomal suspension had a relatively high concentration, because with a lower concentration of liposomes the DSC signal could not be registered. On the other hand, too high concentration of liposomes in the case of spectroscopic measurements was not possible due to the high dispersing properties of the samples.

Line 562-563 The sentence is not understandable and reference [32] is about fluorescence spectroscopy and not DSC.

Response:

Lines 598 in new revised version of the manuscript

A mistake was made in the text - an incorrect literature item was cited. We made the correction and the correct reference was added.

Discussion: the authors should detailed in the method section how the contour plots were built up and the software used.

Response:

Lines 606-607 in new revised version of the manuscript

Contour plots were prepared using procedures included in ORIGIN  program, as written in line 607. This sentence will be added in Methods section in line 606. 

Moreover, the difference between alfa-tocopherol and CT analogue in the interaction with DPPC bilayer should be discussed in more details.

Response:

Discussion regarding comparisons of alpha-tocopherol and CT is now included in Discussion section.

We hope that introduced changes and given explanations fulfil reviewers expectations and that improved manuscript will be acceptable to be published.  

Sincerely,

Authors

Round 2

Reviewer 2 Report

please see file attached
